# Entropy-Based Approach in Selection Exact String-Matching Algorithms

**DOI:** 10.3390/e23010031

**Published:** 2020-12-28

**Authors:** Ivan Markić, Maja Štula, Marija Zorić, Darko Stipaničev

**Affiliations:** 1Faculty of Electrical Engineering, Mechanical Engineering and Naval Architecture, University of Split, 21000 Split, Croatia; 2Department of Electronics and Computing, Faculty of Electrical Engineering, Mechanical Engineering and Naval Architecture, University of Split, 21000 Split, Croatia; maja.stula@fesb.hr (M.Š.); darko.stipanicev@fesb.hr (D.S.); 3IT Department, Faculty of Electrical Engineering, Mechanical Engineering and Naval Architecture, University of Split, 21000 Split, Croatia; marijuki@fesb.hr

**Keywords:** exact string-matching, algorithm efficiency, algorithm performance, entropy, comparison, testing framework

## Abstract

The string-matching paradigm is applied in every computer science and science branch in general. The existence of a plethora of string-matching algorithms makes it hard to choose the best one for any particular case. Expressing, measuring, and testing algorithm efficiency is a challenging task with many potential pitfalls. Algorithm efficiency can be measured based on the usage of different resources. In software engineering, algorithmic productivity is a property of an algorithm execution identified with the computational resources the algorithm consumes. Resource usage in algorithm execution could be determined, and for maximum efficiency, the goal is to minimize resource usage. Guided by the fact that standard measures of algorithm efficiency, such as execution time, directly depend on the number of executed actions. Without touching the problematics of computer power consumption or memory, which also depends on the algorithm type and the techniques used in algorithm development, we have developed a methodology which enables the researchers to choose an efficient algorithm for a specific domain. String searching algorithms efficiency is usually observed independently from the domain texts being searched. This research paper aims to present the idea that algorithm efficiency depends on the properties of searched string and properties of the texts being searched, accompanied by the theoretical analysis of the proposed approach. In the proposed methodology, algorithm efficiency is expressed through character comparison count metrics. The character comparison count metrics is a formal quantitative measure independent of algorithm implementation subtleties and computer platform differences. The model is developed for a particular problem domain by using appropriate domain data (patterns and texts) and provides for a specific domain the ranking of algorithms according to the patterns’ entropy. The proposed approach is limited to on-line exact string-matching problems based on information entropy for a search pattern. Meticulous empirical testing depicts the methodology implementation and purports soundness of the methodology.

## 1. Introduction

String-matching processes are included in applications in many areas, like applications for information retrieval, information analysis, computational biology, multiple variations of practical software implementations in all operating systems, etc. String-matching forms the basis for other computer science fields, and it is one of the most researched areas in theory as well as in practice. An increasing amount and availability of textual data require the development of new approaches and tools to search useful information more effectively from such a large amount of data. Different string-matching algorithms perform better or worse, depending on the application domain, making it hard to choose the best one for any particular case [1,2,3,4,5].

The main reason for analyzing an algorithm is to discover its features and compare them with other algorithms in a similar environment. When features are the focus, the mostly and primarily used parts are time and space resources and researchers want to know how long the implementation of a particular algorithm will run on a specific computer and how much space it will require. The implementation quality and compiler properties, computer architecture, etc., have huge effects on performance. Establishing differences between an algorithm and its implementation features can be challenging [6,7].

An algorithm is efficient if its resource consumption in the process of execution is below some acceptable or desirable level. The algorithm will end execution on an available computer in a reasonable amount of time, or space in efficiently acceptable contexts. Multiple factors can affect an algorithm’s efficiency, such as algorithm implementation, accuracy requirements, and lack of computational power. A few frameworks exist for testing string matching algorithms [8]. Hume and Sunday presented a framework for testing string matching algorithms in 1991. It was developed in the C programming language, and it has been used in the 90′. [9] Faro presented the String Matching Algorithm Research Tool (SMART) framework in 2010 and its improved version six years later. SMART is a framework designed to develop, test, compare, and evaluate string matching algorithms [3].

This paper introduces a model-building methodology for selecting the most efficient string search algorithm, based on a pattern entropy while expressing the algorithm’s efficiency using platform independent metrics. According to their efficiency, the developed methodology for ranking algorithms considers properties of the searched string and properties of the texts that are being searched. This methodology does not depend on algorithm implementation, computer architecture, programming languages specifics, and it provides a way to investigate algorithm strengths and weaknesses. More information about the formal metric definition is described in Section 3.1.3. The paper covers only the fundamental algorithms. Selected algorithms are the basis for the most state-of-the-art algorithms, which belong to the classical approach. More information is described in Section 3.2.2. We also analyzed different approaches, metrics for measuring algorithms’ efficiency, and algorithm types. 

The paper is organized as follows: The basic concepts of string-matching are described in Section 2. Entropy, formal metrics, and related work are described in Section 3 along with the proposed methodology. Experimental results with string search algorithms evaluation models developed according to the proposed methodology are presented in Section 4. Section 5 presents the validation results for the developed models and a discussion. Section 6 concludes this paper.

## 2. String-Matching

String-matching consists of finding all occurrences of a given string in a text. String-matching algorithms are grouped into exact and approximate string-matching algorithms. Exact string-matching does not allow any tolerance, while approximate string-matching allows some tolerance. Further, exact string-matching algorithms are divided into two groups: single pattern matching and multiple pattern matching. These two categories are also divided into software and hardware-based methods. The software-based string- matching algorithms can be divided into character comparison, hashing, bit-parallel, and hybrid approaches. This research focuses on on-line software-based exact string-matching using a character comparison approach. On-line searching means that there is no built data structures in the text. The character-based approach is known as a classical approach that compares characters to solve string-matching problems [10,11].

The character-based approach has two key stages: matching and shift phases. The principle behind algorithms for string comparison covers text scanning with the window of size 𝑚, commonly referred to as the sliding window mechanism (or search window). In the process of comparing the main text *T [1…n]* and a pattern *P [1…m]*, where *m ≤ n*, the aim is to find all occurrences, if any, of the exact pattern 𝑃 in the text 𝑇 (Figure 1). The result of comparing patterns with text is information that they match if they are equal or they mismatch. The length of both windows must be of equal in length, during the comparison phase. First, one must align the window and the text’s left end and then compare the characters from the window with the pattern’s characters. After an exact matching (or mismatch) of pattern with the text, the window is moved to the right. The same procedure repeats until the right end of the window has reached the right end of the text [11,12,13,14,15].

## 3. Methodology

### 3.1. Methodology Description

A state of the art survey shows a lack of platform-independent methodology, which will help choose an algorithm for searching a specific string pattern. The proposed approach for evaluating exact string pattern matching algorithms is formalized in a methodology consisting of six steps, shown in Figure 2, to build a model applicable to data sets and algorithms in a given domain.

The first step of the proposed methodology, shown in Figure 2, is selecting representative texts for domain model building. In the second step, the algorithms are selected. Selected algorithms are limited only to the ones that wanted to be considered. After selecting representative texts for domain model building and algorithms, the searching phase for representative patterns starts in the third step. Representative patterns can be text substrings or the can be randomly created from the domain alphabet. The searching phase means that all representative patterns are searched with algorithms selected in the second step. Search results are collected and expressed in specific metrics. In the fourth step, patterns entropy is calculated. In the fifth step, entropy discretization is applied. Entropy results are discretized and divided into groups by frequency distribution [16,17]. The last step is to classify algorithms in the built model and present the obtained algorithms’ ranking according to the proposed approach.

#### 3.1.1. Representative Patterns Sample Size

The sample size of representative patterns is determined by Equation (1) for finite population [18,19,20]:(1)$$\;n^{\prime }=\frac{n}{1+\frac{z^{2}\times p\left(1-p\right)}{\varepsilon^{2}N}}$$
where *n* is the sample size, *z* is the z-score, *ε* is the margin of error, *N* is the population size, and *p* is the population proportion. The commonly used confidence levels are 90%, 95%, and 99%. Each has its corresponding z-scores provided by tables based on the chosen confidence level (confidence level 0.95 used in the experiment with z-score 1.65). The margin of error means the maximum distance for the sample estimate to deviate from the real value. A population proportion describes a percentage of the value associated with a survey. In theory, we are dealing with an unlimited population since patterns and texts can have an unlimited number of characters. However, in practice, we have to limit populations to a finite number [14,16].

The maximum number of classes in the discretization phase is determined by Equation (2) (n is the total number of observations in the data) [16,17]:(2)$$number\;of\;classes=C=2\times \sqrt[{3}]{n}$$

Also, the range of the data should be calculated by finding minimum and maximum values. The range will be used to determine the class interval or class width. The following Equation (3) is used [16,17]:(3)$$h=\frac{\max\left(values\right)-\min\left(values\right)}{C}$$

#### 3.1.2. Entropy

Shannon entropy is a widely used concept in information theory to convey the amount of information contained in a message. Entropy is a standard measure for the state of order, or better, a disorder of symbols in a sequence. The entropy of a sequence of characters describes the complexity, compressibility, amount of information [21,22,23,24,25].

Suppose that events *A_1_, A_2_, …, A_n_* is defined, and they make a complete set. The following expression is valid $\overset{n}{\underset{i=1}{\sum }}p_{i}=1$, where *p_i_ = p(A_i_)*. Finite system *α* holds all events *A_i_, i = 1, 2, …, n* with probability *p_i_*’s corresponding values. The following form will denote system *α* (Equation (4)) [22]:(4)$$\alpha =\left(\begin{array}{c} A_{1}\\ p_{1} \end{array}\begin{array}{c} A_{2}\\ p_{2} \end{array}\ldots \begin{array}{c} A_{n}\\ p_{n} \end{array}\right)$$

The states of a system α will denote events *A_i_*, *i = 1, 2, …, n*. System *α* is a discrete system with a finite set of states. Every finite system describes some state of uncertainty because it is impossible to know which state is the system in a specific time. The goal is to express quantitatively such uncertainty in some way. It means that a particular function, which will join a specific number to system *α*, should be defined. In that way, the system will have a measure for its uncertainty [22].

The function which quantitatively measures an uncertainty of a system is called *Entropy of system,* and it is defined with the following Equation (5) [22,26]:(5)$$H\left(p_{1},\;p_{2},\;\ldots ,\;p_{n}\right)=-\overset{n}{\underset{i=1}{\sum }}p_{i}logp_{i}$$

The entropy of a system is denoted with *H(α)*. If *p_i_ = 0*, it follows that *p_i_ log p_i_ = 0*. The information theory logarithm base is usually 2, and an entropy unit is called a *bit* (binary digit). The entropy is zero only if one of the probabilities *p_i_ = 1, …, n* is equal *1,* and others are *0*. In that case, there is no uncertainty since it is possible to predict the system’s state precisely. In any other case, entropy is a positive number [22,26].

If the system *α* contains test results, a degree of uncertainty before a test is executed is equal to the entropy of the system *α*. When the test is executed, the degree of uncertainty is zero. The *amount of information* after test execution is larger if the uncertainty was bigger before the test. The information given after the test, denoted with ϑ_α,_ is equal to the entropy of the system *α* (Equation (6)) [22,27]:(6)$$\vartheta_{\alpha }=H\left(\alpha \right)=-\overset{n}{\underset{i=1}{\sum }}p_{i}\log p_{i}=\overset{n}{\underset{i=1}{\sum }}p_{i}(-\log p_{i})$$

Another measure from information theory is Kolmogorov complexity. Although Kolmogorov complexity looks similar to Shannon entropy, they are conceptually different measures. Shannon entropy interprets the smallest number of bits required for the optimal string encoding. Kolmogorov complexity is the minimum number of bits (or the minimum length) from which a particular string can effectively be reconstructed [28,29,30]:

#### 3.1.3. Formal Metric Description

The metrics and the quality attributes that are used for string searching algorithms analysis imply several issues, like the quality of framework (occurs when a quality model does not define a metric), lack of an ontology (occurs when the architectural concepts need quantification), lack of an adequate formalism (when metrics are defined with a formalism that requires a strong mathematical background what causes less metric usability), lack of computational support (occurs when metrics do not produce tools for metrics collection), lack of flexibility (occurs when metrics collection tools are not available in open-source format what causes less ability to modify them) and lack of validation (occurs when cross-validation is not performed). All these issues complicate determining which properties and measures would be useful in selecting metrics and results presentation [31].

Two main approaches exist for expressing the speed of an algorithm. The first approach is formal, analyzing algorithm complexity through algorithm time efficiency (or time complexity, the time required). The second approach is empirical, analyzing particular computer resources usage through space and time efficiency (or space complexity, the memory required). Objective and informative metrics should accompany each approach [2,31,32,33,34,35].

Algorithmic efficiency analysis shows the amount of work that an algorithm will need for execution, and algorithm performance is a feature of hardware that shows how fast the algorithm execution will be done. Formal metrics are usually used for efficiency analysis. A commonly used metric from the formal approach is Big O notation or Landau’s symbol, representing the complexity of an algorithm shown as a function of the input size describing the upper bound for the search time in the worst case. Empirical metrics, like algorithm execution run time usually presented in milliseconds, processor and memory usage, temporary disk usage, long term disk usage, power consumption, etc., are usually used for algorithm performance analysis. The runtime execution metric is difficult to describe analytically, so empirical evaluation is needed through the experiments using execution runtime metrics [4,9,11,14,36,37,38,39,40,41,42].

The proposed methodology focuses on evaluating the speed of algorithms using the character comparisons (CC) metric. CC metric is the number of compared characters of the pattern with the characters of text. Character comparison metric is a measure that is independent of programming language, computational resources, and operating systems, which means that it is platform-independent like formal approaches. However, besides the time complexity, the CC metric covers space complexity in some segments, like the execution run time, and can be programmatically implemented and used like empirical approaches. Thus, this metric is a formal and empirical approach combined [9].

### 3.2. Methodology Implementation

The application of the proposed methodology is presented in the paper for the two domains. For the genome (DNA) domain proposed methodology is implemented and depicted in Figure 3 and for the natural language domain is shown in Figure 4.

The results of entropy calculation (Figure 3 and Figure 4) for each searched pattern are labeled *PattEnt*; the variable *PattEnt* is rounded and labeled *PattEntRound*; the meaning of other variables are *Algo*–selected algorithm, *m*–length of the pattern, *comp*–number of character comparisons per searched pattern. Groups based on the frequency distribution are marked with *PattEntClass*. These designations are used in the following sections in presenting methodology results. Obtained results are used for the algorithm’s ranking list. That is the entropy-based model for selecting the most efficient algorithm for any pattern searched in a particular domain. In the following sections, each step is described in detail.

#### 3.2.1. Selection of Representative Texts for Domain Model Building

For the DNA domain, selected representative texts are the genome data of four different species. For the natural language domain, selected representative texts are English texts from the Canterbury Corpus [43]. The length of a DNA sequence expressed in base pairs (*bp*) varies from a few thousand to several million and even billion *bp*. The DNA character strings are formed with the 4-letter alphabet {A, C, G, T}. The length of texts from the Canterbury Corpus is expressed in bytes and formed of the English alphabet [a-z|A-Z|0–9|!|]. We used the bible subset as the text to be searched because it is more representative of natural English text than the other convenient word lists, and it is publicly released [2,21,44,45].

In detail, the following publicly available representative texts are used for model building:DNA sequences of nucleotides for the DNA domain
⮚*Anabarilius graham* (Kanglang fish; RJVU01051648.1 *Anabarilius grahami* isolate AG-KIZ scaffold371_cov124, whole genome shotgun sequence, 14.747.523 bp (base pairs), 14.3 Mb file size) [46]⮚*Chelonia mydas* (green sea turtle; NW_006571126.1 *Chelonia mydas* unplaced genomic scaffold, CheMyd_1.0 scaffold1, whole genome shotgun sequence, 7.392.783 bp, 7.1 Mb) [47]⮚*Escherichia coli* (NZ_LN874954.1 *Escherichia coli* strain LM33 isolate patient, whole genome shotgun sequence, 49.02.341 bp, 4.8 Mb) [48]⮚*Macaca mulatta* (Rhesus macaque monkey; ML143108.1 *Macaca mulatta* isolate AG07107 chromosome 19 genomic scaffold ScNM3vo_33 × 44 M, whole genome shotgun sequence, 24.310.526 bp, 24.3 Mb) [49]English texts for the natural language domain
⮚Bible (The King James Version, with a size of 4.047.392 bytes) [50]

#### 3.2.2. Selection of Algorithms

Seven commonly used string matching algorithms have been chosen to be ranked with the proposed model: brute force, näive (BF), Boyer-Moore (BM), Knuth Morris Pratt (KMP), Apostolico-Crochemore (AC), quick search (QS), Morris Pratt (MP) and Horspool (HOR) [12,39,51,52,53,54,55,56]. The selected algorithms belong to the group of software-based algorithms that use exact string-matching techniques with a character comparison approach (classical approach) [11]. All selected algorithms used in this experiment match their published version [3,12,39], which might represent the better implementation of the original algorithm [57]. Seven string search algorithms are selected as our baseline for model construction. However, any exact string-matching algorithm that can be evaluated with character comparison metrics can be ranked with the proposed model.

#### 3.2.3. Searching Results for Representative Patterns

For model development, design, and construction in step 3 of the model building (Figure 2), we used 9.725 different patterns. For the DNA domain model, 7.682 patterns are used, and 2.043 patterns of English text from the Canterbury Corpus are used for the natural language domain. The length of patterns ranges from 2 characters to 32 characters.

4.269 patterns for the DNA domain and 1.685 patterns for the natural language domain (or more) are needed to accomplish a confidence level of 95%, that the real value is within ±1% of the surveyed value (Equation (1)). With this confidence level, it can be concluded that a model objectively reflects the modeled domain since a model is constructed with an adequate sample size. 

#### 3.2.4. Patterns Entropy Calculation

Searched patterns are grouped into classes according to their entropy. Entropy is calculated using Equations (5) and (6). For example, for *P = TCGTAACT*, after counting the number of characters in a pattern, *A = 2, C = 2, G = 1, T = 3*, the probabilities respectively are:$$\begin{array}{c} P\left(A\right)=\frac{2}{8}=0.25\;\;P\left(C\right)=\frac{2}{8}=0.25\\ P\left(G\right)=\frac{1}{8}=0.125\;\;\;P\left(T\right)=\frac{3}{8}=0.375 \end{array}$$
$$Entropy=-\left(\frac{2}{8}\times \log_{2}\frac{2}{8}\;\right)-\left(\frac{2}{8}\times \log_{2}\frac{2}{8}\;\right)-\left(\frac{1}{8}\times \log_{2}\frac{1}{8}\;\right)-\left(\frac{3}{8}\times \log_{2}\frac{3}{8}\;\right)\;Entropy=-\left(-0.5\right)-\left(-0.5\right)-\left(-0.375\right)-\left(-0.53064\right)=1.90563906222957\approx 1.91$$

So for a pattern *TCGTAACT* calculated entropy is 1.90563906222957. Entropy for the given pattern from the English text *P = ”e name of the LORD. And the LORD”* is accordingly 3.698391111. Entropies values are rounded to the two decimals (i.e., entropy for pattern *TCGTAACT* is 1.91 and entropy for English text pattern in the above example is 3.7).

#### 3.2.5. Entropies Discretization

The next phase is grouping data into classes or making frequency distribution. Calculated entropies are discretized in classes created by frequency distribution, displaying the number of observations or results in a sample or a given interval. Classes do not need to be represented with the same number of patterns.

Table 1 is just a section of the overall patterns entropy classification for the DNA domain.

For the DNA domain model the total number of observations in the data *n* = *91.* The observation data are distinct values of calculated entropies rounded up two decimals, for DNA domain there are totally 91 items (0 | 0.34 | 0.53 | 0.54 | 0.70 | 0.81 | 0.90 | 0.95 | 0.99 | … | 1.96 | 1.97 | 1.98 | 1.99 | 2.00). The number of classes for the DNA domain, applying Equation (2) is 9, width of classes after applying Equation (3) is 0.22. The Table 2 shows entropy classes after discretization with the number of patterns in each of them.

Table 3 is just a section of the overall patterns entropy classification for the natural language domain.

For the natural language domain, the total number of observations in the data *n = 105*. The observation data for natural language domain are (0 | 1 | 1.5 | 2 | 2.16 | 2.25 | 2.41 | 2.5 | 2.62 | … | 3.83 | 3.84 | 3.86 | 3.87 | 3.88), totally 105 items. The number of classes for the natural language domain, applying Equation (2) is 9, the width of classes after applying Equation (3) is 0.46. The Table 4 shows entropy classes for the natural language domain after discretization with the number of patterns in each of them.

Entropy classes containing a small number of patterns affect the model the least since such patterns are rare and occur in less than 0.5% of cases. Examples of such patterns are TTTTTTTTCTTTTTTT, AAAGAAA, and LL. When a pattern does not belong to any entropy class, the relevant class is the first closest entropy class. 

## 4. Classification of Algorithms in the Built Model

The algorithm analysis results integrated into a model, provide a ranking list of algorithms by their efficiency, measured with character comparison metrics, correlated with the searched pattern entropy. More efficient algorithms perform fewer characters comparison when finding a pattern in a text. The model proposes a more efficient algorithm for string matching for a given pattern based on the entropy class to which the observed pattern belongs.

The results presented in Table 5 and Table 6 give a ranking list of the selected algorithms grouped by the entropy class. The percentages shown in the result tables represent a proportion of pattern searching results for a particular algorithm, which might be smaller or greater than the remaining algorithms inside the quartile. For example, in Table 5, if the searched pattern belongs to the entropy class 1 (number of representative patterns is 9), 55.88% of the searching results for a given entropy class with the QS algorithm are in the first quartile, 14.71% are in the second quartile, 29.41% are in the third quartile (Figure 5). When patterns are searched with the BM algorithm, 47.92% of the searching results expressed as CC count is in the first quartile, 23.53% are in the second quartile, and 25% are in the third, and 8.33% are in the fourth quartile. In this case, for a given pattern, the built model suggests using the QS algorithm as the most efficient algorithm. The selected algorithm is considered an optimal algorithm that will make fewer character comparisons (CC) than others for most patterns being searched belonging to the entropy class 1.

Entropy classes in Table 5 are defined in Table 2.

In Table 5, for the entropy class 8 (number of representative patterns searched is 1451), the model shows that the BM algorithm is the most efficient. In 61.95% of cases for patterns in the entropy class 8, the BM algorithm made the least characters comparison versus the other six algorithms evaluated with the model. In 24.38% cases, BM was second best; in 13.68% cases was the third and never was the worse.

Entropy classes in Table 6 are defined in Table 4.

In Table 6, for example, for the entropy class 6 (number of representative patterns searched is 393), the model shows that the QS algorithm is the most efficient. In 70.13% of cases for patterns in the entropy class 6, the QS algorithm made the least characters comparison versus the other six algorithms evaluated with the model. In 29.87% of cases, QS was second best and never was the worse. For the entropy class 7 (number of representative patterns searched is 283), the model shows that the most efficient is the BM algorithm. In 65.02% of cases for patterns in the entropy class 7, the BM algorithm made the least characters comparison versus the other six algorithms evaluated with the model. In 34.98% of cases, BM was second best and never was the worse.

## 5. Methodology Validation and Discussion

For model validation, the seventh and ninth entropy classes (961 and 4692 patterns) were selected for the DNA domain, and the sixth (393 patterns) and ninth classes (221 patterns) were selected for the natural language domain. The model classes chosen for validation have the highest number of representative patterns and are characteristic for the specific domains.

The selected patterns for validation are not part of the patterns set with which the model was created. For the DNA domain model, also a different text is chosen for validation. The DNA domain model is validated with the DNA sequence *Homo sapiens* isolate HG00514 chromosome 9 genomic scaffold HS_NIOH_CHR9_SCAFFOLD_1, whole genome shotgun sequence, 43.213.237 bp, 39 Mb as the text [58]. The natural language domain is validated with the natural language text set from the Canterbury Corpus. [43]

Before the model validation process, a check was made to see if the selected patterns were sufficiently representative for model validation. The check was done with the central limit theorem. The set of patterns used in the validation phase has a normal distribution (Figure 6, Mean = 1.900, and Std. Dev = 0.064) as a set of patterns used in model building (Figure 7, Mean = 1.901, and Std. Dev = 0.059), which means that patterns used to validate the model represent a domain.

Other entropy classes of patterns discretized character comparisons also follow the normal distribution. The basis in the model validation phase is to verify if the test results differ from the developed model results presented in Table 5 and Table 6.

For comparing the two data sets (model results and test results), double-scaled Euclid distance, and Pearson correlation coefficient were used.

Double-scaled Euclidian distance normalizes raw Euclidian distance into a range of 0–1, where 1 represents the maximum discrepancy between the two variables. The first step in comparing two datasets with double-scaled Euclidian methods is to compute the maximum possible squared discrepancy (*md*) per variable *i* of *v* variables, where *v* is the number of observed variables in the data set. The *md_i_ = (Maximum for variable i-Minimum for variable i)*^2^, where 0 (0%) is used for minimum and 1 (100%) for maximum values for double-scaled Euclid distance. The second step’s goal is to produce the scaled variable Euclidean distance, where the sum of squared discrepancies per variable is divided by the maximum possible discrepancy for that variable, Equation (7):(7)$$d_{1}=\sqrt{\overset{v}{\underset{i=1}{\sum }}\left(\frac{{\left(p_{1i}-p_{2i}\right)}^{2}}{md_{i}}\right)}$$

The final step is dividing scaled Euclidian distance with the root of *v*, where *v* is the number of observed variables, Equation (8). Double-scaled Euclid distance easily turns into a measure of similarity by subtracting it from 1.0. [16,59,60,61,62]:(8)$$d_{2}=\frac{\sqrt{\sum_{i=1}^{v}\left(\frac{{\left(p_{1i}-p_{2i}\right)}^{2}}{md_{i}}\right)}}{\sqrt{v}}$$

Table 7 shows the usage of the double-scaled Euclidian distance method for entropy class 7 of DNA

Applying Equation (8) on Table 7, column “Scaled Euclidean (*d_1_*)”, gives a double-scaled Euclidian distance of 0.227. Subtracting double-scaled Euclidian Distance from 1 gives a similarity coefficient of 0.773 or 77%.

Table 8 shows the results of the calculated double-scaled Euclid distance and corresponding similarity coefficient.

Converting double-scaled Euclidian distance to a context of similarity, it is possible to conclude that the built model is similar to the validation results with a high degree of similarity. The seventh and ninth classes from the built model for the DNA domain have a similarity coefficient with their validation results of 77%. The high percentage of similarity also has the sixth and ninth classes from the built model for the natural language domain with their validation results of 80% and 86%. The results for validated classes obtained in the validation process are extremely similar to the results from the built model. A proportion of searched pattern character comparisons for a particular algorithm inside the quartile is similar to the built model.

Pearson’s correlation coefficient is used to check the correlation between data from the model and data from the validation phase. Pearson correlation coefficients per classes are shown in Table 9.

The seventh and ninth classes from the built model for the DNA domain have a linear Pearson’s correlation coefficient with their validation results. The sixth and ninth classes from the natural language domain’s built model have a linear Pearson’s correlation coefficient with their validation results. Pearson’s correlation coefficient shown in Figure 8 indicate that the values from the built model (*x*-axis, Model) and their corresponding validation result (*y*-axis, validation) follow each other with a strong positive relationship.

Using the double-scaled Euclidean distance in the validation process shows a strong similarity between the built model and validation results. In addition to the similarity, a strong positive relationship exists between classes selected from the built model and validation results proven by Pearson’s correlation coefficient. Presented results show that it is possible to use the proposed methodology to build a domain model for selecting an optimal algorithm for the exact string matching. Except for optimal algorithm selection for a specific domain, this methodology can be used to improve the efficiency of string- matching algorithms in the context of performance, which is in correlation with empirical measurements.

The data used to build and validate the model can be downloaded from the website [63].

## 6. Conclusions

Proposed methodology for ranking algorithms is based on properties of the searched string and properties of the texts being searched. Searched strings are classified according to the pattern entropy. This methodology is expressing algorithms efficiency using platform independent metrics thus not depending on algorithm implementation, computer architecture or programming languages characteristics. This work focuses on classical software-based algorithms that use exact string-matching techniques with a character comparison approach. For any other type of algorithms, this methodology cannot be used. The used character comparisons metrics is platform-independent in the context of formal approaches, but the number of comparisons directly affects the time needed for algorithm execution and usage of computational resource. Studying the methodology, complexity, and limitations of all available algorithms is a complicated and long-term task. The paper discusses, in detail, available metrics for string searching algorithms properties evaluation and proposing a methodology for building a domain model for selecting an optimal string searching algorithm. The methodology is based on presenting exact string-matching results to express algorithm efficiency regardless of query pattern length and dataset size. We considered the number of compared characters of each algorithm expressed by the searched string entropy for our baseline analysis. High degrees of similarity and a strong correlation between the validation results and the built model data have been proven, making this methodology a useful tool that can help researchers choose an efficient string- matching algorithm according to the needs and choose a suitable programming environment for developing new algorithms. Everything that is needed is a pattern from a specific domain by which the model is built, and the model will suggest using the most optimal algorithm for usage. The defined model finally selects the algorithm that will most likely run up the least character comparison count in pattern matching. This research does not intend to evaluate the algorithm logic and programming environment in any way; the main reason for comparing the results of algorithms is the construction of the algorithm selection model. The built model is straightforwardly extendable with other algorithms; all required is adequate training data sets. Further research is directed to find additional string characteristics, besides pattern entropy, that can enhance developed methodology precision for selecting more efficient string search algorithms.

## Figures and Tables

**Figure 1 entropy-23-00031-f001:**
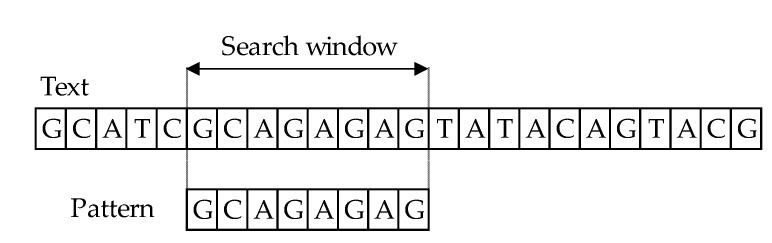
Exact string matching.

**Figure 2 entropy-23-00031-f002:**
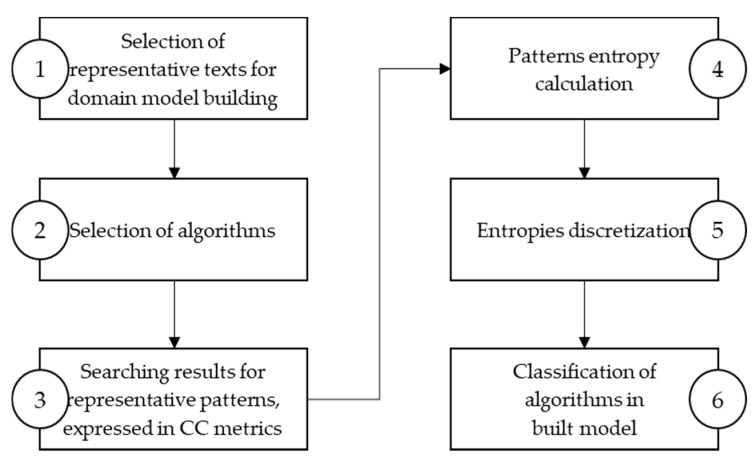
Methodology for building model based on the entropy approach for string search algorithms selection.

**Figure 3 entropy-23-00031-f003:**
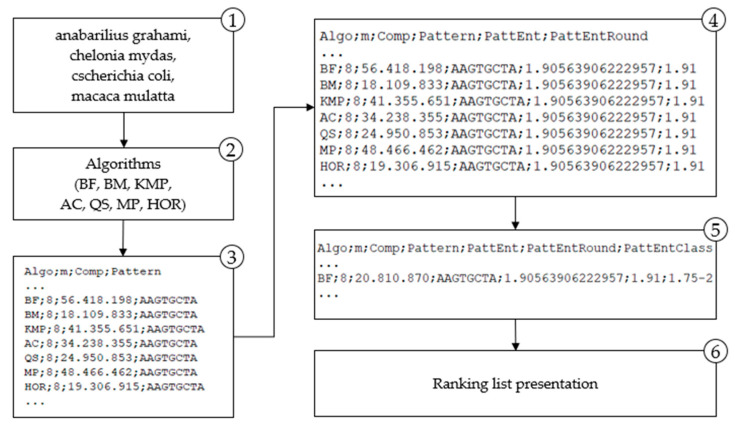
Methodology implementation for the DNA domain.

**Figure 4 entropy-23-00031-f004:**
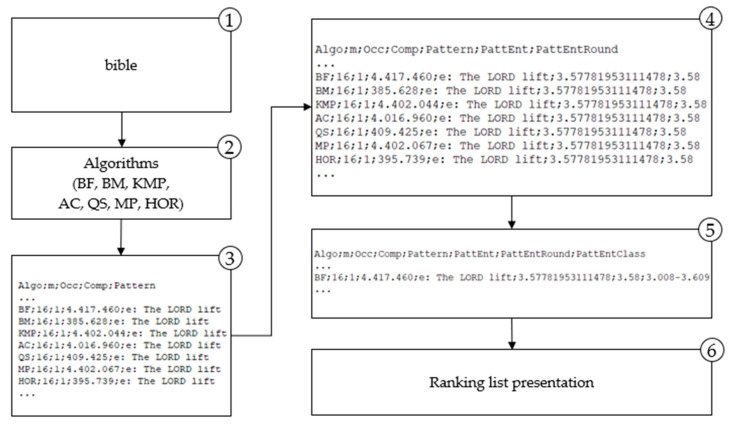
Methodology implementation for the natural language domain.

**Figure 5 entropy-23-00031-f005:**
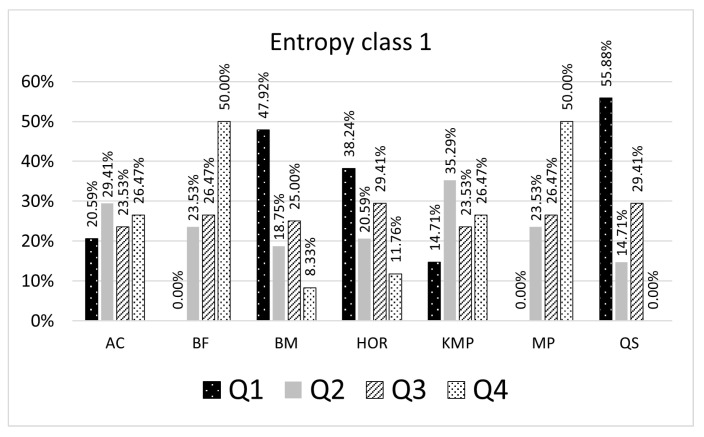
Algorithms ranking for entropy class 1.

**Figure 6 entropy-23-00031-f006:**
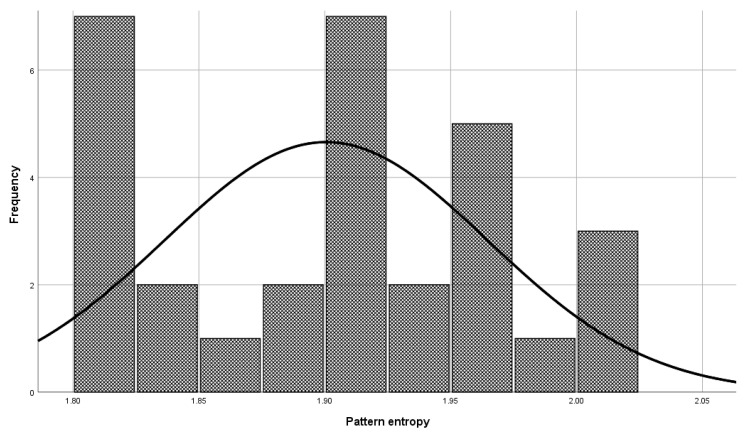
Patterns used in the validation phase for the entropy class 9 of the DNA domain.

**Figure 7 entropy-23-00031-f007:**
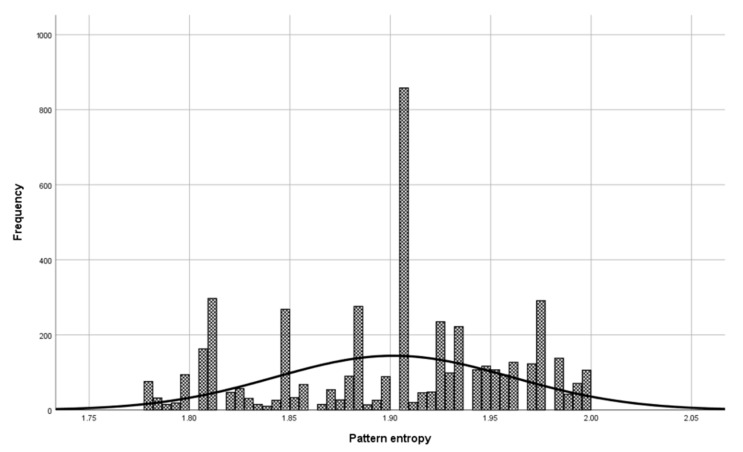
Patterns used in model building for the entropy class 9 of the DNA domain.

**Figure 8 entropy-23-00031-f008:**
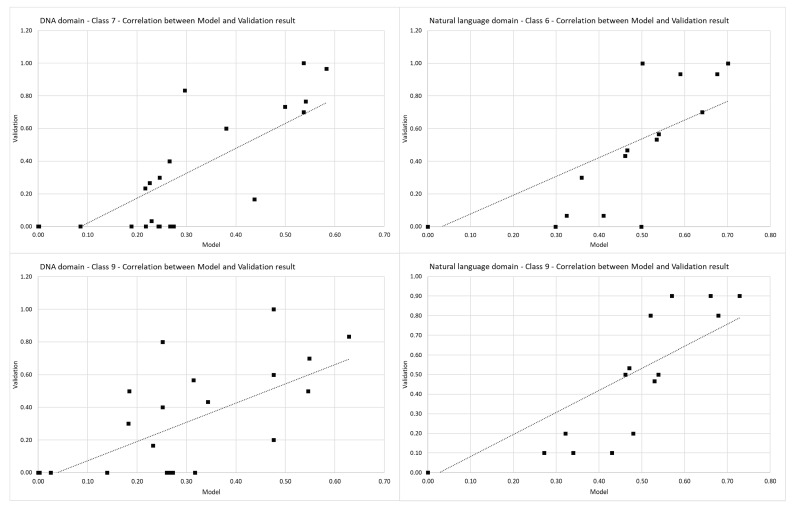
Pearson’s linear correlation coefficient.

**Table 1 entropy-23-00031-t001:** Data discretization for the DNA domain.

Pattern	PattEnt	PattEntRound	PattEntClass
AAAAAAAA	0.0000000000000	0.00	<0.22222
TTTTTTTTCTTTTTTT	0.3372900666170	0.34	0.22222–0.44444
AACAAAAA	0.5435644431996	0.54	0.44444–0.66667
AAAAAAACAAACAACA	0.6962122601251	0.70	0.66667–0.88889
TGGTAAAAAAAAAAAA	1.0612781244591	1.06	0.88889–1.11111
AAAAAGCG	1.2987949406954	1.30	1.11111–1.33333
CAAG	1.5000000000000	1.50	1.33333–1.55556
CCTACTAAACACCGTA	1.7640976555739	1.76	1.55556–1.77778
GCATACCTTTCGCAGC	1.9362781244591	1.94	≥1.77778

**Table 2 entropy-23-00031-t002:** Entropy classes after discretization for the DNA domain.

Class No.	Entropy Class	Number of Patterns
1	<0.22222	9
2	0.22222–0.44444	1
3	0.44444–0.66667	23
4	0.66667–0.88889	72
5	0.88889–1.11111	195
6	1.11111–1.33333	278
7	1.33333–1.55556	961
8	1.55556–1.77778	1.451
9	≥1.77778	4.692
	Total	7.682

**Table 3 entropy-23-00031-t003:** Data discretization for the natural language domain.

Pattern	PattEnt	PattEntRound	PattEntClass
Mm	0.0000000000000	0.00	<0.46004
We	1.0000000000000	1.00	0.92007–1.38011
Full	1.5000000000000	1.50	1.38011–1.84014
off from	2.1556390622295	2.16	1.84014–2.30018
ine enemies, eve	2.6556390622295	2.66	2.30018–2.76021
Joseph remembere	3.0243974703477	3.02	2.76021–3.22025
e: The LORD lift	3.5778195311147	3.58	3.22025–3.68028
they have, and deliver our lives	3.7508072359050	3.75	≥3.68028

**Table 4 entropy-23-00031-t004:** Entropy classes after discretization for the natural language domain.

Class No.	Entropy Class	Number of Patterns
1	<0.46004	3
2	0.46004–0.92007	0
3	0.92007–1.38011	151
4	1.38011–1.84014	53
5	1.84014–2.30018	383
6	2.30018–2.76021	393
7	2.76021–3.22025	283
8	3.22025–3.68028	556
9	≥3.68028	221
	Total	2.043

**Table 5 entropy-23-00031-t005:** Algorithms ranking model for DNA texts and patterns.

Entropy Class/Algorithm	1	2	3	4	5	6	7	8	9
Quartile 1									
AC	20.59%	0.00%	20.00%	12.61%	5.74%	8.99%	8.51%	5.43%	2.53%
BF	0.00%	0.00%	0.00%	0.00%	0.00%	0.00%	0.00%	0.00%	0.00%
BM	47.92%	100.00%	56.67%	51.82%	57.20%	61.11%	58.26%	61.95%	62.83%
HOR	38.24%	0.00%	46.67%	49.55%	49.88%	51.06%	49.92%	53.92%	54.60%
KMP	14.71%	0.00%	0.00%	0.00%	0.00%	0.00%	0.00%	0.00%	0.00%
MP	0.00%	0.00%	0.00%	0.00%	0.00%	0.00%	0.00%	0.00%	0.00%
QS	55.88%	100.00%	53.33%	51.35%	55.36%	53.97%	54.08%	53.68%	54.85%
Quartile 2									
AC	29.41%	100.00%	16.67%	35.14%	41.15%	31.22%	38.03%	36.47%	34.34%
BF	23.53%	0.00%	13.33%	22.97%	20.95%	18.52%	21.75%	21.58%	26.36%
BM	18.75%	0.00%	43.33%	28.38%	24.28%	29.63%	22.92%	24.38%	23.21%
HOR	20.59%	100.00%	30.00%	14.41%	20.95%	29.89%	22.54%	24.32%	18.41%
KMP	35.29%	0.00%	13.33%	26.58%	20.95%	18.52%	26.65%	21.58%	27.22%
MP	23.53%	0.00%	13.33%	22.97%	20.95%	18.52%	21.75%	21.58%	27.22%
QS	14.71%	0.00%	43.33%	23.42%	25.94%	28.57%	21.63%	25.08%	18.25%
Quartile 3									
AC	23.53%	0.00%	53.33%	25.23%	29.18%	36.24%	26.54%	31.67%	31.41%
BF	26.47%	0.00%	16.67%	24.77%	27.43%	25.66%	24.56%	24.36%	26.01%
BM	25.00%	0.00%	0.00%	19.80%	16.05%	9.26%	18.82%	13.68%	13.95%
HOR	29.41%	0.00%	23.33%	35.59%	26.18%	19.05%	27.38%	21.77%	26.72%
KMP	23.53%	100.00%	43.33%	21.17%	34.66%	41.53%	29.63%	37.77%	25.15%
MP	26.47%	0.00%	33.33%	24.77%	27.43%	25.66%	24.56%	24.56%	25.15%
QS	29.41%	0.00%	3.33%	25.23%	15.96%	17.46%	24.28%	21.24%	26.65%
Quartile 4									
AC	26.47%	0.00%	10.00%	27.03%	23.94%	23.54%	26.93%	26.43%	31.72%
BF	50.00%	100.00%	70.00%	52.25%	51.62%	55.82%	53.69%	54.06%	47.63%
BM	8.33%	0.00%	0.00%	0.00%	2.47%	0.00%	0.00%	0.00%	0.01%
HOR	11.76%	0.00%	0.00%	0.45%	2.99%	0.00%	0.17%	0.00%	0.27%
KMP	26.47%	0.00%	43.33%	52.25%	44.39%	39.95%	43.72%	40.65%	47.63%
MP	50.00%	100.00%	53.33%	52.25%	51.62%	55.82%	53.69%	53.87%	47.63%
QS	0.00%	0.00%	0.00%	0.00%	2.74%	0.00%	0.00%	0.00%	0.25%

**Table 6 entropy-23-00031-t006:** Algorithms ranking model for the natural language texts and patterns.

Entropy Class/Algorithm	1	2	3	4	5	6	7	8	9
Quartile 1									
AC	0.00%	0.00%	0.00%	0.00%	0.00%	0.00%	0.00%	0.00%	0.00%
BF	0.00%	0.00%	0.00%	0.00%	0.00%	0.00%	0.00%	0.00%	0.00%
BM	66.67%	0.00%	40.56%	60.38%	57.18%	64.05%	65.02%	58.81%	66.06%
HOR	33.33%	0.00%	35.06%	30.19%	38.72%	41.01%	51.24%	56.47%	52.04%
KMP	0.00%	0.00%	0.00%	0.00%	0.00%	0.00%	0.00%	0.00%	0.00%
MP	0.00%	0.00%	0.00%	0.00%	0.00%	0.00%	0.00%	0.00%	0.00%
QS	100.00%	0.00%	100.00%	84.91%	79.23%	70.13%	59.01%	59.71%	57.01%
Quartile 2									
AC	100.00%	0.00%	51.67%	50.94%	50.00%	50.13%	58.66%	50.18%	72.85%
BF	0.00%	0.00%	0.00%	0.00%	0.00%	0.00%	0.00%	0.00%	0.00%
BM	33.33%	0.00%	59.44%	39.62%	42.82%	35.95%	34.98%	41.19%	33.94%
HOR	66.67%	0.00%	64.94%	69.81%	61.28%	58.99%	48.76%	43.53%	47.96%
KMP	100.00%	0.00%	0.00%	0.00%	0.00%	0.00%	0.00%	0.00%	0.00%
MP	0.00%	0.00%	0.00%	0.00%	0.00%	0.00%	0.00%	0.00%	0.00%
QS	0.00%	0.00%	0.00%	15.09%	20.77%	29.87%	40.99%	40.29%	42.99%
Quartile 3									
AC	0.00%	0.00%	48.33%	49.06%	50.00%	49.87%	41.34%	49.82%	27.15%
BF	0.00%	0.00%	36.67%	33.96%	33.08%	32.41%	31.45%	33.45%	32.13%
BM	0.00%	0.00%	0.00%	0.00%	0.00%	0.00%	0.00%	0.00%	0.00%
HOR	0.00%	0.00%	0.00%	0.00%	0.00%	0.00%	0.00%	0.00%	0.00%
KMP	0.00%	0.00%	44.44%	49.06%	45.90%	46.58%	47.70%	46.04%	47.06%
MP	33.33%	0.00%	44.44%	41.51%	45.90%	46.08%	45.94%	45.50%	46.15%
QS	0.00%	0.00%	0.00%	0.00%	0.00%	0.00%	0.00%	0.00%	0.00%
Quartile 4									
AC	0.00%	0.00%	0.00%	0.00%	0.00%	0.00%	0.00%	0.00%	0.00%
BF	100.00%	0.00%	63.33%	66.04%	66.92%	67.59%	68.55%	66.55%	67.87%
BM	0.00%	0.00%	0.00%	0.00%	0.00%	0.00%	0.00%	0.00%	0.00%
HOR	0.00%	0.00%	0.00%	0.00%	0.00%	0.00%	0.00%	0.00%	0.00%
KMP	0.00%	0.00%	55.56%	50.94%	54.10%	53.42%	52.30%	53.96%	52.94%
MP	66.67%	0.00%	55.56%	58.49%	54.10%	53.92%	54.06%	54.50%	53.85%
QS	0.00%	0.00%	0.00%	0.00%	0.00%	0.00%	0.00%	0.00%	0.00%

**Table 7 entropy-23-00031-t007:** Example of usage double-scaled Euclidian distance on DNA entropy class 7.

*i*	Built Model (p1)	Validation (p2)	Scaled Euclidean (d_1_)
1	0.58263	0.96667	0.14749
2	0.22916	0.03333	0.03835
…			
28	0.43718	0.16667	0.07318

**Table 8 entropy-23-00031-t008:** Double-scaled Euclid distance for DNA and natural language classes.

Model vs. Validation Result for:	DNA		Natural Language	
	Double-Scaled Euclidean Distance	Similarity Coefficient	Double-Scaled Euclidean Distance	Similarity Coefficient
Class 6			0.194	0.806 (80%)
Class 7	0.227	0.773 (77%)		
Class 9	0.231	0.769 (77%)	0.145	0.855 (86%)

**Table 9 entropy-23-00031-t009:** Pearson correlation coefficient for DNA and natural language classes.

Model vs. Validation Result for:	DNA	Natural Language
Class 6		0.848
Class 7	0.795	
Class 9	0.685	0.905

## Data Availability

Data is contained within the Appendix A.

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
