# Peer review of "Entropy-Based Approach in Selection Exact String-Matching Algorithms"

_entropy, 2020, doi:10.3390/e23010031_

Round 1
Reviewer 1 Report
There are some general problems in this paper. First, the connection of the cited reference with the corresponding context is imprecise. For example, in L54, it cites [7]. However, ref. [7] is a general text book for algorithms that has more than 1000 pages, and there is no close relation to the “SMART framework,” as described in the context. Similarly, I don’t see why it cites ref. [5] at L44 and L76.
Second, the explanations of some statistical formulas are unclear or possibly with some errors. For example, in L111, both $n$ and $N$ are population sizes. What are their differences? In eq. (7) at L345, the upper limit of the sum should be $v$, not $n$. The description of eq. (7) is so general and has no direct connection to the computation of Table 7. I mean, the correspondence between the items (e.g. variables, index i) of eq. (7) with the validation data sets (e.g. algorithms).
As a conclusion, I think the authors can try to clarify these points in order to make the paper easier to be understood.
Author Response
Ivan Markić
Vukovarska 3
88320 Ljubuški
Bosna i Hercegovina
Split, 19.12.2020
Submission: Reply to reviewers
Dear editors,
Dear reviewers,
We appreciate the comments on our paper and the opportunity to revise our work. In the following, we explain the revisions that we made according to the reviewers’ comments. Our reply is always in blue.
We hope that our revision meets the requirements of the reviewers and the journal. Please contact Ivan Markić (ivan.markic@outlook.com) for any further correspondence about the paper.
Please see the attachment.
Sincerely,
Ivan Markić, Maja Štula, Marija Zorić, Darko Stipaničev

Reviewer 2 Report
This paper presents a quasi-review of mathing algorithms for string correspondence. It's not too clear what the real noveltie against the current state-of-the-art is. This should be clearly addressed from the very beginning at the Abstract and Introduction.
My second major concern is: there is plenty of new trend approaches focused on deep learning techniques for classificators and matchers of many sort of data. Despite the fact that authors state they do not pretend to evaluate nothing to do with computation, at least this aspect should be discussed with more depth.
Other comments:
Cites to references in boxes [] are always presented after a full stop when it should be done before the sentence ends. Please check this format issues.
Sect 1.1 provides specific details, so that it does not make too much sense appearing within "Introduction" section.
Links to publicly available databases must be included.
Sections have to be reformulated. Some results are presented within subsections of Sect 2.
Lines 247-250. These should be listed as normal equations. Inline equations like here do not fit well with the alignement.
Author Response

(The authors gave the same response as above.)

Round 2
Reviewer 2 Report
Major comments have been sufficiently responsed.